# 3D Printing of Dietary Products for the Management of Inborn Errors of Intermediary Metabolism in Pediatric Populations

**DOI:** 10.3390/nu16010061

**Published:** 2023-12-25

**Authors:** Paola Carou-Senra, Lucía Rodríguez-Pombo, Einés Monteagudo-Vilavedra, Atheer Awad, Carmen Alvarez-Lorenzo, Abdul W. Basit, Alvaro Goyanes, María L. Couce

**Affiliations:** 1Departamento de Farmacología, Farmacia y Tecnología Farmacéutica, I+D Farma (GI-1645), Facultad de Farmacia, Materials Institute (iMATUS) and Health Research Institute of Santiago de Compostela (IDIS), Universidade de Santiago de Compostela, 15782 Santiago de Compostela, Spain; paola.carou@rai.usc.es (P.C.-S.); lucia.rodriguez.pombo@rai.usc.es (L.R.-P.); carmen.alvarez.lorenzo@usc.es (C.A.-L.); 2Servicio de Neonatología, Unidad de Diagnóstico y Tratamiento de Enfermedades Metabólicas Congénitas, Health Research Institute of Santiago de Compostela (IDIS), Hospital Clínico Universitario de Santiago de Compostela, Universidad de Santiago de Compostela, RICORS, CIBERER, MetabERN, 15706 Santiago de Compostela, Spain; eines.monteagudo.vilavedra@sergas.es; 3Department of Clinical, Pharmaceutical and Biological Sciences, University of Hertfordshire, College Lane, Hatfield AL10 9AB, UK; a.awad@herts.ac.uk; 4Department of Pharmaceutics, UCL School of Pharmacy, University College London, 29-39 Brunswick Square, London WC1N 1AX, UK; a.basit@ucl.ac.uk; 5FABRX Ltd., Henwood House, Henwood, Ashford, Kent TN24 8DH, UK; 6FABRX Artificial Intelligence, 27543 O Saviñao, Spain

**Keywords:** chewable formulations and oral drug products, dietary therapy and supplements, intermediary metabolic diseases, extrusion-based three-dimensional printing of personalized pharma-inks, pediatric patients, direct ink writing 3D-printed drug delivery systems, on-demand dispensing of pharmaceuticals and medicines

## Abstract

The incidence of Inborn Error of Intermediary Metabolism (IEiM) diseases may be low, yet collectively, they impact approximately 6–10% of the global population, primarily affecting children. Precise treatment doses and strict adherence to prescribed diet and pharmacological treatment regimens are imperative to avert metabolic disturbances in patients. However, the existing dietary and pharmacological products suffer from poor palatability, posing challenges to patient adherence. Furthermore, frequent dose adjustments contingent on age and drug blood levels further complicate treatment. Semi-solid extrusion (SSE) 3D printing technology is currently under assessment as a pioneering method for crafting customized chewable dosage forms, surmounting the primary limitations prevalent in present therapies. This method offers a spectrum of advantages, including the flexibility to tailor patient-specific doses, excipients, and organoleptic properties. These elements are pivotal in ensuring the treatment’s efficacy, safety, and adherence. This comprehensive review presents the current landscape of available dietary products, diagnostic methods, therapeutic monitoring, and the latest advancements in SSE technology. It highlights the rationale underpinning their adoption while addressing regulatory aspects imperative for their seamless integration into clinical practice.

## 1. Introduction

The term “Inherited Metabolic Disorder” (IMD) encompasses a diverse array of genetic disorders, where a deficiency in a specific enzyme, transporter, or regulatory protein disrupts normal metabolic pathways [1,2]. This enzymatic deficiency impedes the degradation of natural endogenous substrates, leading to their accumulation in various body tissues. Without appropriate treatment, these disorders can have fatal consequences. Given their genetic origin, IMDs are relatively rare, typically manifesting in children who display early symptoms, underscoring the critical importance of prompt diagnosis for timely intervention and the prevention of metabolic or severe multisystemic consequences. The estimated global birth prevalence of IMD stands between 50 and 125 per 100,000 live births [3], making them a significant contributor to pediatric mortality and morbidity worldwide. Although individually rare, collectively, these disorders represent a common health concern.

Inborn errors of intermediary metabolism (IEiM) constitute a substantial portion of IMDs, affecting the breakdown of low-molecular-weight nutrient compounds. This group includes 13 out of 24 categories in the current International Classification of IMD [1]. Main disorders within IEiM result from genetic defects in enzymes or cofactors involved in the metabolism of amino acids (e.g., phenylketonuria, maple syrup urine disease (MSUD), homocystinuria, tyrosinemias, organic acidemias, urea cycle disorders (UCDs)), carbohydrates (e.g., galactosemia, hereditary fructose intolerance, glycogen storage disease), and fatty acids (e.g., fatty acid β-oxidation defects) [4]. Disruptions in these pathways, often caused by enzyme alterations, can lead to toxic substance accumulation or energy production deficiency, which is detectable through specific biochemical markers. Newborn screening (NBS), implemented worldwide since the 1960s, has facilitated early diagnosis and treatment of IEiM [5,6,7].

IEiMs typically manifest as multisystemic diseases with both neurological and non-neurological symptoms, occasionally accompanied by distinctive physical features [8]. Characterized by a symptom-free neonatal period, signs of intoxication emerge during early childhood due to toxic compound accumulation. However, manifestations can also occur later, displaying intermittent, chronic, or progressive patterns leading to neurodegeneration [9]. Despite extensive efforts, the first-line treatment remains dietary control and nutritional supplementation. Administering treatment poses challenges for both patients and their caregivers, with doses often being prescribed on a trial-and-error basis.

This review aims to offer an encompassing perspective on current dietary product therapies for pediatric IEiM patients, highlighting their advantages and disadvantages. Additionally, it explores recent innovations in three-dimensional (3D) printing technology, emphasizing its potential in personalizing pediatric medication for improved adherence, palatability, and dose customization. The use of biosensors for early diagnosis and drug monitoring is also examined. Artificial intelligence (AI) is proposed as a supportive tool for dose predictions and 3D printing performance. Finally, future trends related to regulatory aspects for implementing 3D printing in clinical practice are addressed.

## 2. From Catalysts to Cures: Conventional Metabolic Therapies

The primary objective of therapy for IMD is to restore metabolic homeostasis while minimizing the detrimental effects of the interruption [10]. Traditional approaches to managing these disorders center on reinstating the balance between substrate and product. This involves primarily decreasing the substrate through dietary restrictions, increasing the product via dietary supplementation, or enhancing the conversion from substrate to product through enzyme or cofactor replacement (Figure 1).

### 2.1. Substrate Reduction

Reducing the substrate aims to limit the availability of compounds that cannot be fully metabolized by the defective enzyme, cofactor, or regulatory protein [10]. The accumulation of such compounds may lead to the formation of secondary byproducts, which are sometimes toxic, resulting in clinical manifestations. Limiting the substrate helps restore a steady-state balance in the pathway, which is achievable through various means, which will be discussed in more detail as follows.

#### 2.1.1. Substrate Reduction Therapy (SRT)

Substrate reduction therapy (SRT) aims to reduce the synthesis of substrates to a level that can be effectively cleared by the impaired enzyme, essentially blocking their excessive production (Figure 1a). This therapeutic approach is particularly notable in lysosomal storage diseases (LSDs), which arise from metabolic deficiencies in lysosomal hydrolases, which are responsible for breaking down macromolecular lipids and carbohydrates [11]. SRT is also employed in other metabolic disorders such as aminoacidopathies, organic acidurias, or UCD [10,12]. An example of this approach involves the use of 2-(2-nitro-4-trifluoro-methylbenzoyl)-1,3-cyclohexanedione (NTBC), also known as nitisinone, in treating tyrosinemia type 1 [13]. NTBC functions by inhibiting the activity of 4-hydroxyphenylpy-ruvate dioxygenase, preventing the formation of toxic metabolites. When combined with a tyrosine diet restriction, NTBC is currently employed for the management of tyrosinemia type I, improving the prognosis and reducing the necessity for liver transplantation.

#### 2.1.2. Substrate Dietary Restrictions

The approach of substrate dietary restrictions involves limiting specific substrates in the diet and providing deficient products or alternative energy sources (Figure 1b). Treatment typically entails a lifelong restriction of the intake of certain toxic substances by limiting the amounts of natural protein, sugars, or lipids in the diet. This is combined with the administration of medical foods and/or supplements that are deficient. An exemplary application of this approach is the dietary treatment for phenylketonuria (PKU) [14]. In PKU, elevated plasma levels of phenylalanine, caused by defective phenylalanine hydroxylase (PAH) (i.e., converts phenylalanine into tyrosine) lead to severe cognitive impairment and psychiatric disability. To mitigate this, a phenylalanine-restricted diet is established, supplemented with a phenylalanine-free and tyrosine-rich amino acid formula [15].

#### 2.1.3. Scavenger Therapy

Many issues associated with metabolic disorders stem from toxic metabolites produced by the transformation of accumulated substrate. Scavenger therapy is a medical approach aimed at removing these toxic metabolites from the bloodstream (Figure 1c) [16,17].

In UCD, high levels of ammonia accumulate, causing hyperammonemia, which can lead to brain injury. Dietary restrictions and supplementation may not always suffice to maintain low ammonia levels. Here, scavenger therapy involves the intake of sodium benzoate or sodium phenylbutyrate, commonly used as ammonia scavengers, to facilitate the urinary excretion of nitrogen, thus reducing ammonia levels in the bloodstream [17].

### 2.2. Providing the Product

In addressing IEiMs, alternative treatments focus on augmenting the production of the necessary product, either by facilitating the conversion of the substrate to the desired product or by directly supplying the product (Figure 1f) [18].

#### 2.2.1. Cofactor Supplementation

Cofactors, comprising organic or inorganic molecules or metallic ions, enhance enzyme function for maximal catalytic efficiency or endurance (Figure 1d) [19]. Some IMDs show clinical improvement with cofactor supplementation. For instance, MSUD, an aminoacidopathy, exhibits various degrees of partial enzyme activity, including the intermediate thiamine-responsive phenotype. In such cases, thiamine supplementation enhances enzyme activity [20]. Another example is sapropterin dihydrochloride, also known as tetrahydrobiopterin (BH4), used in treating patients with PKU [21]. BH4 acts as a cofactor for PAH. By enhancing the enzyme’s activity, it aids in the oxidative metabolism of phenylalanine in two ways: either by reducing or maintaining blood amino acid concentrations within the target therapeutic range and/or by increasing natural protein or phenylalanine tolerance. Nevertheless, it has been observed that only approximately 20–30% of patients exhibit responsiveness to sapropterin.

#### 2.2.2. Enzyme Replacement

Enzyme Replacement Therapy (ERT) involves providing an exogenous source of the deficient enzyme to catalyze substrates (Figure 1e) [21]. This approach is utilized in managing PKU. A commercially available drug, pegvaliase (marketed as Palynziq^®^), a PEGylated bacterial phenylalanine ammonia lyase, substitutes the deficient enzyme in PKU, reducing phenylalanine levels [22].

#### 2.2.3. Dietary Supplementation

In IEiM, the primary challenge lies in the inadequate production of products within affected pathways. Insufficient products trigger physiological perturbations, necessitating dietary supplementation for homeostasis control in most diseases (Figure 1g) [23]. Assessing nutritional therapy products may reveal a justified need for a pharmaceutical or dietary product [24]. As a result of dietary treatment, patients often require supplementation for normal cellular function. Unfortunately, there is a lack of products tailored to meet these needs, and pharmaceutical compounding at the point-of-care (PoC) becomes the predominant alternative (i.e., capsules/powders dispersed with water or food) for administering treatments. This approach is exemplified in the PKU, where the conversion of phenylalanine into tyrosine is affected. Tyrosine plays an important role in synthesizing various molecules, including epinephrine, dopamine, or melanin, making it an essential intermediary amino acid for body functioning. As such, PKU treatment includes specific formulas enriched with tyrosine to address PAH deficiency and maintain normal tyrosine levels.

The long-term prognosis depends on factors such as the age at diagnosis, the type of IEiM, and adherence to dietary treatment [25]. Slight dietary modifications or events such as infections or surgeries can sometimes trigger major episodes of metabolic decompensation, emphasizing the dynamic nature of patient requirements over time [26,27]. Maintaining motivation for adherence to diet throughout life is challenging yet essential. Unfortunately, on a restricted diet, concentrations of essential amino acids often fall below the normal range, and adherence to the diet is frequently inadequate [28,29].

Another application of this method involves the use of dietary supplements to manage long-chain fatty acid oxidation disorders (LC-FAOD) [30]. The inclusion of medium-chain triglyceride (MCT) supplements is pivotal in treating these patients. Apart from dietary restrictions on long-chain triglycerides, patients require supplementation with MCTs or triheptanoin, constituting at least 10% of their total calorie intake. The specific supplementation needs and proportions vary according to the individual’s clinical condition, posing a significant challenge in ensuring adherence to the prescribed diet.

### 2.3. Liver Transplantation

Liver transplantation emerges as another therapeutic avenue for treating certain IEiMs [31,32], particularly in cases with severe phenotypes (Figure 1e). The transplanted liver assists in correcting the accumulation of harmful metabolites by restoring enzymatic activity. Depending on the disease’s pathology and the presence of extrahepatic manifestations, liver transplantation can either offer a curative solution or enhance clinical stability, allowing patients to follow a less restrictive diet. By averting episodes of severe decompensation, liver transplantation also contributes to preventing neurological damage.

### 2.4. Gene Therapy Research

Given that IEiMs are predominantly monogenic diseases, gene therapy stands out as a potential treatment avenue (Figure 1e) [33]. This involves delivering therapeutic genes using innocuous vectors. Adeno-associated viral vectors are commonly chosen for liver-related gene therapies due to their safety and efficacy, while lentiviral and retroviral vectors are explored for central nervous system diseases [19]. Multiple strategies, including ex vivo or in vivo gene therapy and other approaches utilizing antisense oligonucleotides, ribonucleic acid interference (RNAi), or delivering mRNA or microRNA with nanoparticles are under development [34,35,36,37]. The advent of the CRISPR/Cas9 system has furthered exploration for some IEiMs, such as UCD, organic acidurias, MSUD, PKU, or tyrosinemia type 1, among others, showing promising results for in vivo gene-editing procedures [38,39].

## 3. Navigating Metabolic Mazes: Pioneering Precision Medicine in IEiM Management

The constraints of traditional treatments outlined in the preceding section underscore the pressing need for a novel approach to the management of IEiM. The inherent variability within the pediatric population, even among individuals with the same disease, necessitates a tailored approach for each patient [40]. The emergence of “precision medicine” or “personalized medicine” represents a promising alternative to the limitations of the conventional “one-size-fits-all” model, particularly in pediatric populations [41]. This innovative paradigm focuses on factors influencing individual therapeutic responses, including genetic profiles, medical conditions, and the inherent properties of active ingredients, ultimately aiming to enhance treatment efficacy [42]. Figure 2 illustrates the virtuous cycle of personalized medicine within this evolving healthcare model, where the unique needs of children affected by IEiM are considered comprehensively, including strategies for detecting, treating, and monitoring—all of which could be tailored to a patient’s unique profile [43].

### 3.1. Detection and Monitoring

Early detection plays a pivotal role in enhancing the quality of life for those with IEiM. The utilization of blood spotted and dried on a matrix, known as “dried blood spot” or DBS has revolutionized neonatal screening and early diagnosis, surpassing conventional blood testing [5]. Current biomarkers’ analysis methods, predominantly performed in centralized laboratories, rely on techniques such as liquid chromatography (LC) or gas chromatography (GC) coupled with mass spectrometry (MS), enabling quantitative assays with low detection limits. In DBS, blood samples are obtained through finger pricking using a lancet [44]. Following clear instructions provided by the clinician and after adequate training, patients or caregivers can independently draw blood droplets. Subsequently, a drop of blood is applied to a sampling paper, which is dried and posted to a laboratory. Thereafter, the blood spots are extracted from the paper and the concentration of the analyte is quantified [45]. The DBS test, implemented as part of routine clinical practice in hospitals, offers advantages such as the ease of non-invasive blood collection, the ability for patients to obtain samples at home, painless procedures, low blood volumes, and the testing of various analytes, including proteins, lipids, and small organic or non-organic molecules [46,47]. However, the equipment used is expensive, and sensitivity and specificity are specific to each molecule depending on where the cut-off is set. Another drawback includes the use of small volumes and potential interference from hemoglobin [48,49].

The DBS test extends beyond detection and can be applied to therapeutic dietary and/or pharmacological monitoring (TD/FM), facilitating dose adjustments based on active substance levels [50]. While TD/FM is crucial for optimizing outcomes and avoiding adverse effects, current methods have limitations, including the need for skilled operators, patient discomfort, and the inability to monitor dynamic drug level changes [51]. Wearable or portable sensors have emerged as a dynamic solution for real-time and non-invasive therapeutic drug monitoring [43]. These sensors, measuring drugs in biological fluids such as saliva, capillary blood, tears, or sweat, enable physicians to correlate pharmacokinetic profiles with optimal outcomes. Numerous biosensors have been explored for drug monitoring across various drugs, demonstrating their potential for personalized medicine.

Biosensors have found significant applications in managing liver glycogen storage diseases (GSDs), a group of inherited disorders affecting glycogen metabolism and regulation [52]. These disorders primarily affect the liver, leading to hypoglycaemia caused by irregular hepatic glycogen degradation and glucose release. The treatment typically involves administering slow-release forms of glucose ‘like uncooked cornstarch’. It should be noted that life-threatening hypoglycaemia is a risk in patients with GSDs, particularly overnight. Continuous glucose monitors (CGMs), akin to those used in diabetes mellitus patients, have been recently explored [53,54]. These CGMs track glucose levels in the interstitial fluid, providing crucial 24 h glucose data with multiple daily readings and early predictive warnings for impending hypoglycaemia. Comprising a disposable unit inserted under the skin and a reusable transmitter that sends glucose data to a receiver (e.g., mobile phone), these sensors, in conjunction with dietary adjustments, have proven to be a safe and effective means of optimizing treatment, thus advancing towards personalized medicine.

Other examples of such include touch-based sensors for monitoring levodopa in sweat [55], electrochemical aptamer-based sensors for measuring vancomycin in plasma [56], wearable electroactive sensors for the quantification of paracetamol in saliva and sweat [57], wearable glove-embedded sensors for the monitoring of paracetamol and paroxetine in sweat [58], microneedle electrochemical sensors that analyze fentanyl concentrations [59], flexible vinyl terephthalate substrates with carbon nanotube-modified working electrodes for the detection of caffeine in sweat [60], ring-based sensors for the simultaneous detection of tetrahydrocannabinol and alcohol in saliva [61], and eyeglasses for the detection of alcohol in tears [62].

These affordable biosensors utilize electrochemical (based on aptamers, antibodies, enzymes, or ions) [63], optical (colorimetric, Raman scattering signals or fluorescence) [64], or electromagnetic [65] principles (Figure 3). Offering continuous measurement of drug or biomarker concentrations, biosensors are sensitive enough to detect minimal changes over extended periods, providing vital data for optimal therapeutic effects and exemplifying the promise of personalized medicine in IEiM management.

### 3.2. Advanced Therapies through 3D Printing Technology

First-line therapies for most metabolic disorders commonly involve dietary restrictions and supplementation with essential amino acids to prevent deficiencies [10]. Acknowledging the variability in patient responses, there has been a shift towards precision nutrition, recognizing the individual variability in patients and their differing requirements [71]. The concept of precision nutrition considers interindividual factors such as age, blood levels, and specific patient responses [68,72,73], which are particularly crucial in pediatric patients where treatment imbalances can yield pronounced responses [74]. However, current manufacturing models, which are reliant on large drug batches with limited design flexibility, often necessitate time-consuming and unstable extemporaneous formulations [75].

Despite ongoing development and testing of various strategies, including gene therapy, dietary modifications or restrictions remain fundamental therapeutic tools. However, managing patients with these rare diseases necessitates comprehensive care involving multidisciplinary teams and extensive involvement of families and caregivers and necessitates comprehensive care involving education, support, and adherence to dietary treatments [76,77]. Challenges persist, including the complexity of nutritional calculations, limited food product labeling information, poor palatability of supplements, and dosage fragmentation issues [25]. Addressing these challenges may involve leveraging AI techniques, biosensors, and the development of mobile applications to enhance patient education and metabolic monitoring, potentially reducing hospital visits and improving patient autonomy [78,79].

To circumvent these challenges, novel nutritional strategies have emerged in recent years, particularly the development of “medical foods” for patients who cannot be managed with standard diet modifications [80]. However, access to such treatments remains limited, and the perception of dietary products designed for children as anything but a regular diet persists. Hence, the imperative to enhance various aspects of these nutritional treatments and explore new technologies for their design has become evident.

In recent years, 3D printing technologies have garnered substantial attention across various industries [81,82], including medicine, owing to their capacity to create 3D objects layer-by-layer from digital models [83]. Particularly within the pharmaceutical realm, 3D printing offers unique opportunities to fabricate customized pharmaceutical products [84,85], providing full design customization, flexibility, and high control over drug release, dose personalization, and the capacity to print multiple drugs [86,87,88,89].

From a technical approach, the American Society for Testing and Materials (ASTM) standards classify 3D printing technologies into seven categories: binder jetting, directed energy deposition, material extrusion, material jetting, powder bed fusion, sheet lamination, and vat photopolymerization. The process involves initial product design using computer-aided design (CAD), which is followed by slicing the design into a G-code, instructing the printer to create the desired model object using various materials, termed pharma-inks (e.g., filament, binder solution, paste), which depend on the technology employed (Figure 4).

Semi-solid extrusion (SSE) is a material extrusion technique, depositing gels, waxes, or pastes to create solid objects without requiring high temperatures, thereby preventing drug degradation and ensuring rapid, low-temperature printing [90]. SSE has facilitated the development of tailored 3D printed dosage forms (termed printlets) and drug delivery systems such as orodispersible printlets [91,92,93,94,95], polypills (i.e., multidrug-loaded printlets) [88,96,97], suppositories [98,99,100,101], immediate [102,103] and controlled release tablets [104,105], microneedles [106], intravesical inserts [107], or ophthalmologic [108] and dermal [109] patches (Figure 5).

#### Precision Medicine for Children: SSE 3D Printing’s Tailored Solutions

The availability of licensed pharmaceuticals for pediatric use significantly lags behind those for adults, resulting in a dearth of safe formulations with accurate doses that children are both capable and willing to consume [110]. Designing an ideal formulation for this demographic necessitates several considerations: (i) minimal impact on their lifestyle; (ii) individualized dosing to avoid potential treatment toxicity or inefficacy; (iii) non-toxic excipients in formulations; (iv) convenient and reliable administration; and (v) palatability and ensuring patient compliance [111]. Conventional solid and liquid dosage forms still present limitations, especially concerning swallowing difficulties inherent in solid forms for pediatric use, as well as stability and dosing errors in liquid forms. Consequently, novel oral formulations such as chewable formulations and orodispersible tablets and films have gained attention due to their ease of administration, safety, and stability. Leveraging 3D printing techniques, particularly SSE, facilitates the personalization of these innovative pharmaceutical forms, enabling variations in shapes, sizes, flavors, and individualized doses to enhance patient compliance and treatment safety.

Orodispersible films (ODFs), administered without water, significantly enhance patient acceptability. Utilizing SSE 3D printing for ODF production circumvents swallowing difficulties while tailoring the dose to suit individual patient needs [91,112]. Additionally, ODFs serve as a potential alternative for pediatric psychiatric patients, as they cannot remove the films from their mouths, thereby reducing the risk of choking. Building on these advantages, warfarin-containing films were developed using SSE, showcasing the potential of printing technologies in producing on-demand, patient-specific doses of this drug. SSE 3D printing was also employed to create orodispersible tablets, aimed at mitigating the challenge of swallowing [113]. To this end, hydrochlorothiazide printlets were manufactured as pediatric formulations, simplifying the process, preventing drug degradation by not reaching high temperatures, and ensuring the desired therapeutic outcome. Furthermore, the choice of excipients used in these formulations was suitable for children. Additionally, the feasibility of using minitablets and minicaplets tailored for pediatric patients was explored, providing an alternative to conventional oral solid dosage forms with sizes conducive to easy swallowing [114,115]. Although a wide range of dosage forms can be developed using SSE, the most promising ones to be implemented in clinical practice are chewable tablets. This conclusion is based on results obtained from a single-site, two-part survey that involved children aged between 4 and 11 years, wherein 79% of the children have expressed preference towards this pharmaceutical form [116]. The application of SSE 3D printing has been transformative, particularly in the development of chewable formulations tailored for pediatric populations. These chewable formulations, characterized by their versatility, offer an alternative delivery system for children for whom swallowing conventional tablets is challenging [117]. Additionally, as they are chewed and broken down into smaller pieces, they can be produced in sizes larger than conventional tablets, allowing higher amounts of drugs to be loaded into them [118]. Chewable formulations share some advantages with conventional tablets, including portability and long-term stability [119]. Utilizing pharmaceutical excipients, these formulations address sensory characteristics with coloring agents, sweeteners, flavors, and design aesthetics, creating shapes that are appealing to children [118]. The SSE technology has successfully developed various chewable formulations with different shapes, colors, flavors, and textures, showing high acceptability among pediatric patients [120,121,122,123,124,125,126,127,128]. Moreover, formulations containing multi-drugs or supplements (termed polypills) could be successfully prepared in a simple and easy manner using SSE, which highlights the versatility of the 3D printing technology [123]. Some examples of 3D-printed chewable formulations are shown in Figure 6.

Remarkably, SSE 3D printing’s potential was showcased in a clinical trial aimed at treating children affected by the rare metabolic disease maple syrup urine disease (MSUD) [74]. The study was the first to demonstrate that SSE 3D printing offers a feasible, rapid, and automated approach for developing tailored dosage forms in a hospital setting, enabling the creation of medicines with good acceptability among pediatric patients. Personalized chewable printlets containing isoleucine in varying flavors and colors were developed (Figure 6b), demonstrating tighter control over blood levels compared to conventional capsules. The positive responses from children and caregivers indicate the acceptability of these flavored printlets, marking a significant stride in tailored treatment. These findings were in agreement with a previous acceptability study that showed that chewable printlets are more appealing compared to other types of pharmaceutical dosage forms [116]. Following these advantages, SSE 3D printing was successfully employed to develop chewable tablets with personalized doses of amlodipine besylate (Figure 6c) [120] and propranolol hydrochloride (Figure 6a) [129], tailored to meet the needs and preferences of pediatric patients.

SSE 3D printing has emerged as an effective approach to producing tailored medicines, offering not only specific dose requirements but also improved acceptability among pediatric patients by aligning the formulation with their preferences for flavor, texture, and color. A diverse array of excipients can be utilized, including pectin, gelatin, carrageenan, chocolate, cellulose derivatives, and various sugars as sucrose substitutes, among others. This technology allows for the incorporation of colorants and flavors, enabling the adaptation of medication to suit individual patient preferences. Moreover, the production of small-sized formulations such as minitablets or minicaplets addresses swallowing difficulties, particularly in children. When formulated with appropriate excipients, these chewable formulations represent promising candidates for pediatric treatment, improving adherence and therapeutic response [130].

Recent advancements have introduced cereal-based 3D-printed dosage forms tailored for the pediatric population [131]. Concealing drugs within food and beverages is a common practice to aid medication intake among pediatric patients in hospitals. However, this often involves manually crushing tablets or opening capsules before mixing them with food or liquid, which can be time-consuming and prone to dosing errors. In a recent study, crushed cereal was employed as the ink for SSE 3D printing of oral formulations [131]. These formulations, available in various shapes, such as a letter, star, heart, torus, and flower, contained ibuprofen and paracetamol.

The technology’s cost-effectiveness, safety, and capacity to create personalized medicines in hospital settings underscore its potential as a groundbreaking method for pediatric drug delivery and nutritional products [74].

### 3.3. Artificial Intelligence (AI) in Therapeutics

AI denotes computational technologies emulating human intelligence mechanisms encompassing thought, deep learning, adaptation, engagement, and sensory understanding [132]. The integration of AI in medicine, starting from the 1950s, has gained significant momentum in recent years due to advancements in modern computers and the vast volume of digital data accessible for collection, interpretation, and application in healthcare [133]. Clinical data can be overwhelming, but AI technologies can adeptly manage and uncover hidden information within this medical big data [134]. By enabling pattern recognition within intricate datasets, AI can facilitate dose predictions using algorithms, continuously from newly generated data, and progressively aiding in working more “smartly” [135]. Due to that, AI is currently being exploited to simplify and optimize various processes, improving the efficiency and effectiveness of individual tasks. Consequently, AI can be used for the digitalization of industries.

Machine learning (ML), a branch of AI, is instrumental in furnishing machines with consciousness, employing algorithms to learn from complex datasets, which is increasingly applicable in the pharmaceutical industry for expediting drug development processes [135]. ML’s utility extends to virtual screening predictions [136,137], diagnosis [138], and drug efficacy [139], as well as predictions concerning drug absorption, distribution, metabolism, excretion, and toxicity (ADMET) [140]. Real-time data from biosensors further empower ML models to make dose adjustments and clinical decisions, thereby enhancing treatment performance, ensuring safety, and averting adverse effects [43].

ML models have also found utility in optimizing 3D printing performance, particularly in the context of versatile and flexible 3D printing technologies, notably SSE. The multifaceted nature of these technologies offers numerous possibilities in product design, dose, composition, and printing parameters, usually requiring expert navigation [141]. The integration of ML with 3D printing technologies can help guide the printing process, providing the user with the ideal printing parameters to be used (e.g., temperature, nozzle diameter, rheological properties of the pharma-ink). However, it is not limited to that. In fact, AI’s integration in printing technologies, such as fused deposition modeling (FDM) [142,143], digital light processing (DLP) [144], and selective laser sintering (SLS) [145], allows for the prediction of printing outcomes and dissolutions rates of the 3D printed dosage forms in addition to the critical manufacturing parameters.

For instance, M3DISEEN, an open-source software, serves as a guide for drug development via FDM printing, predicting crucial manufacturing parameters [146]. Similarly, ML models have been used to predict the printability of pharma-inks in inkjet printing, enabling the adjustment of doses for patients exhibiting varying treatment responses [147]. Additionally, ML models have been developed for the design and fabrication of microneedles, showcasing AI’s role in predicting complex geometries for innovative medical devices [148].

Advanced ML systems are pivotal in promoting technical enhancement, expediting drug discovery and development, and facilitating the integration of novel high-potential technologies within the medical domain. AI continues to revolutionize therapeutic development, providing insights and efficiencies unattainable through traditional approaches.

## 4. Regulatory and Financial Challenges

Specifically for treating metabolic rare diseases, the adoption of low-scale, small-batch manufacturing using 3D printing technologies can supplant the preparation of extemporaneous formulations offering pharmaceutical compounding, offering a more personalized approach in hospital settings [74]. The integration of 3D printing technology within hospitals is progressing towards point-of-care manufacturing models, with ongoing research emphasizing the potential of these techniques in formulation development and patient care. However, due to their novelty and rapid advancement, there exists a lack of a comprehensive regulatory framework, with no specific guidelines dedicated to the development of printed pharmaceutical forms. Limited clinical studies have been conducted using 3D printing technologies, specifically SSE [74,149,150]. Recently, a study tested placebo printlets in six human subjects [151]. For the first time, magnetic resonance imaging was used to assess the in vivo disintegration of printlets. This innovation underlines the ongoing efforts to gather evidence and data on 3D-printed dosage forms, aiming to facilitate the practical implementation of this technology in clinical practice.

In 2017, the United States Food and Drug Administration (FDA) developed a guidance that included technical considerations specifically for devices produced using printing technologies [152]. This guide outlined technical considerations and recommended testing and characterization procedures for printed devices. While several printed medical devices have received FDA approval, the only pharmaceutical product currently available in the market produced by 3D printing is Spritam^®^ [153]. It remains ambiguous whether regulations will apply solely to the final pharmaceutical product or if they will encompass the various steps and components involved in the entire process.

Nevertheless, the potential and promising outcomes of 3D printing technologies, in both hardware and software, have enabled hospital-based 3D printing, leading to reduced processing times and costs and a stride towards more accessible personalized medicine. This progress has prompted governmental actions aimed at introducing new regulatory frameworks and altering policies concerning these new pharmaceutical products. For instance, the Medicines and Healthcare Products Regulatory Agency (MHRA) has proposed a regulatory framework for products produced at the PoC concerning medicine approvals, clinical trials, and regulatory compliance evaluations, enhancing the safety and effectiveness of medicine manufacturing at PoC while adapting regulatory requirements [154]. Additionally, the FDA has published a paper addressing feedback on this topic, evaluating the existing risk-based regulatory framework applicable to PoC manufacturing, identifying challenges, and proposing future policy initiatives in this domain [155].

In terms of cost, 3D printing presents a mix of cost challenges and advantages in pharmaceutical manufacturing. The cost structure relies on several factors, starting with the availability of various 3D printers on the market, where the price range for pharmaceutical 3D printers is typically reasonable. Comparatively, tableting machines, the traditional counterparts, also come with considerable costs. Moreover, the raw materials used in 3D printing—similar to those used in conventional tablet manufacturing—do not incur additional expenses [89,119,156]. Furthermore, the majority of developed formulations generally eliminate the need for extra post-printing steps, thus not incurring added costs. Pre-printing steps of SSE pharma-inks involve simple mixing and heating at low temperatures, followed by cooling to solidify the formulation before printing, again not adding significantly to costs [90]. Contrastingly, traditional methods such as tableting and capsule filling often require granulation, adding extra expenses. The cost of time is another essential factor to consider in pharmaceutical manufacturing. While 3D printers are generally slower compared to traditional tableting machines, the latter excels in mass production, ensuring rapidity and efficiency. However, when it comes to personalization, the speed advantage of tableting diminishes significantly. Achieving customization through tableting requires individual molds for each variant, resulting in increased costs and time investment. Conversely, 3D printing offers a distinct advantage in this regard. Despite being slower than tableting for mass production, 3D printing allows swift alterations in designs without incurring additional expenses [157]. This flexibility is particularly beneficial in tailored medication production, offering a cost-effective means to adjust formulations and doses to meet individual patient needs.

## 5. Quality Control Assays

The incorporation of 3D printing in pharmaceuticals necessitates demonstrating efficacy and safety in a clinical trial setting, which is a challenge requiring approval from competent authorities. Regulatory agencies mandate a comprehensive dataset to ensure participation, making the application of 3D printing in clinical trials complex. Ensuring the quality of produced dosage forms is paramount, involving quality control (QC) tests such as mass uniformity, content uniformity, dissolution performance, impurity control, and stability.

Challenges arise in conducting quality control tests, especially for chewable tablets, a promising form for pediatric patients affected by IEiM disorders, due to the lack of specific monographs in major pharmacopeias [119]. Although FDA recommendations exist [158], conducting these conventional QC tests can be impractical for on-demand production at clinical trial sites using 3D printing due to their destructive, laborious, and costly nature. Process Analytical Technologies (PATs) have recently been explored in 3D printing to overcome these limitations [159], including in-line mass uniformity via a balance-3D printer system, expediting processes, and saving time. Additionally, near-infrared (NIR) spectroscopy has shown promise as an alternative to destructive techniques for quantifying drug loading in 3D-printed dosage forms [157,160,161,162]. Exploring further non-destructive methods and integrating them seamlessly into the 3D printing systems for automated QC remains a crucial area of ongoing research.

## 6. Conclusions

In the realm of treating children affected by IEiM, the focus for several decades has centered on dietary therapy and nutritional supplementation, with exploratory avenues like gene therapy or technological editing gradually emerging, but they are yet to be implemented in clinical practice.

Given the vast variability and heterogeneity inherent in these metabolic disorders, a personalized approach becomes imperative to tailor dietary and pharmacological treatments to each patient’s unique profile. Novel strategies have been proposed to address this necessity by revolutionizing the detection, treatment, and monitoring of IEiM. For instance, the integration of DBS tests into routine clinical practices has enabled early diagnosis and monitoring of several IEiMs. This minimally invasive assay facilitates various analyte tests and blood collection, thereby providing a convenient diagnostic method.

Moreover, the adoption of SSE printing technology offers a potential alternative for manufacturing personalized medicines. SSE’s distinctive characteristics, particularly in producing chewable formulations, present opportunities for customizing IEiM therapies. Properties like shape, dosage, and sensory attributes (e.g., flavor, smell, color, texture) can be tailored to suit individual patient preferences, enhancing the acceptability of dietary products and pharmacological therapies.

In conjunction with this, the utilization of AI, particularly ML, in tandem with biosensors emerges as a supportive tool in optimizing the development of these therapies. ML models serve as prediction tools for drug performance, screening, and diagnosis. Additionally, ML assists in guiding the printing process by predicting printability and drug dissolution rates, thereby streamlining drug development in a cost-effective manner.

While these innovative strategies, complemented by AI, offer promising avenues for pediatric therapy, the absence of a robust regulatory framework, especially in the 3D printing of pharmaceutical or dietary products, remains a significant challenge. Future endeavors are essential in the regulatory domain to facilitate the implementation of these methodologies within clinical practice. Collaborative efforts are warranted to bridge this gap, ensuring the safe and effective integration of these transformative techniques into pediatric healthcare.

## Figures and Tables

**Figure 1 nutrients-16-00061-f001:**
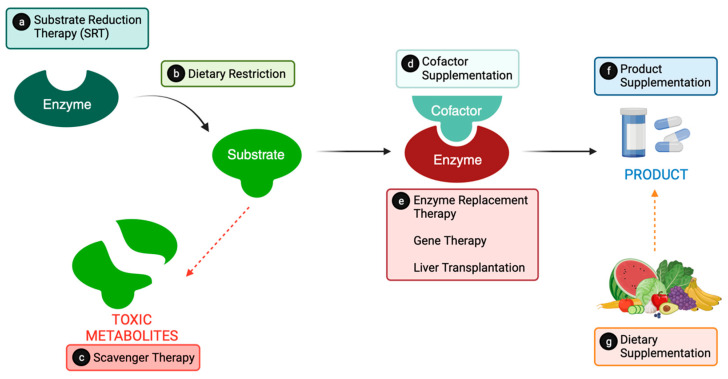
Schematic representation of an enzymatic metabolic pathway alongside established and emerging treatments for Inherited Metabolic Disorders: (**a**) Substrate Reduction Therapy (SRT); (**b**) Dietary Restriction; (**c**) Scavenger Therapy; (**d**) Cofactor Supplementation; (**e**) Enzyme Replacement Therapy, Gene Therapy and Liver Transplantation; (**f**) Product Supplementation; and (**g**) Dietary Supplementation.

**Figure 2 nutrients-16-00061-f002:**
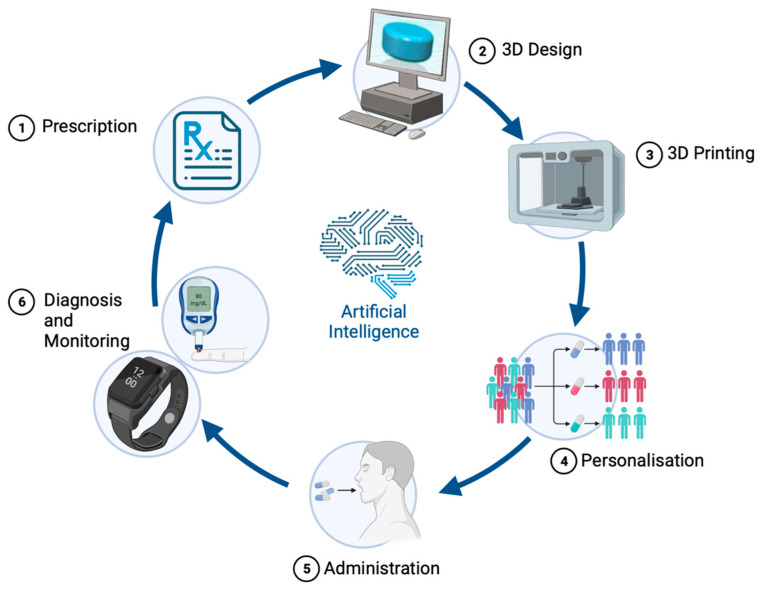
The iterative process of the virtuous cycle of personalized medicine: (1) a clinician prescribes a tailored dose based on the patient’s disease state; (2) utilizing computer-aided design software, a suitable dosage form is digitally modeled in 3D; (3) the design is transmitted to a 3D printer situated in a hospital or pharmacy; (4) personalized medication meeting specific patient needs and preferences (e.g., dosage, form, flavor, aesthetics) is manufactured and dispensed; (5) the medication is administered to the patient; and, finally, (6) remote monitoring of drug blood levels and performance facilitates continuous adjustments in the treatment plan to achieve optimal therapeutic outcomes. Artificial intelligence is integrated throughout the entire process to streamline operations.

**Figure 3 nutrients-16-00061-f003:**
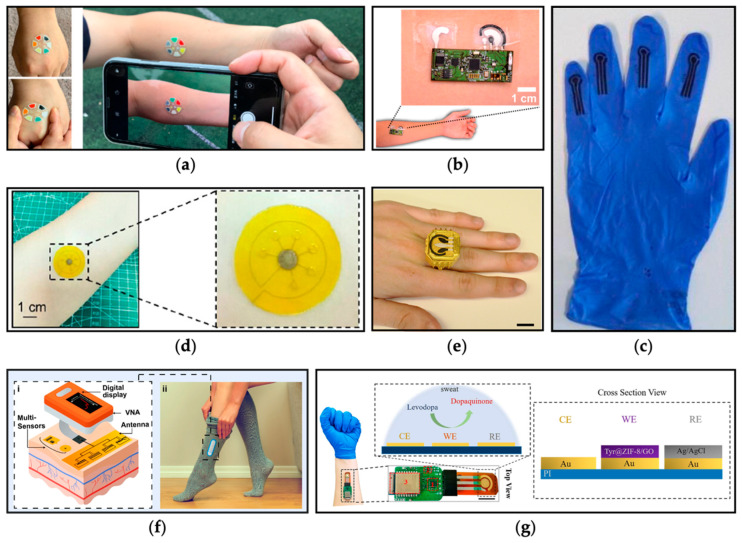
Examples of wearable biosensors: (**a**) colorimetric sensing patch detecting pH, calcium, and chloride ions in sweat [66]; (**b**) iontophoretic-biosensing temporary tattoo for ethanol quantification in sweat [67]; (**c**) electrochemical wearable glove-embedded sensors measuring uric acid, paracetamol, paroxetine, and ethinylestradiol in sweat [58]; (**d**) surface-enhanced Raman scattering (SERS) sensor monitoring paracetamol in sweat [68]; (**e**) electrochemical ring sensor simultaneously measuring tetrahydrocannabinol and alcohol in saliva (scale bar: 15 mm) [61]; (**f**) electromagnetic wave sensing technology for glucose monitoring: (**i**) schematic representation of the biosensor; and (**ii**) the sensor is worn within the sock apparatus [69]; and (**g**) electrochemical sensor for levodopa detection in sweat [70]. All figures were reprinted with permission from their original sources.

**Figure 4 nutrients-16-00061-f004:**
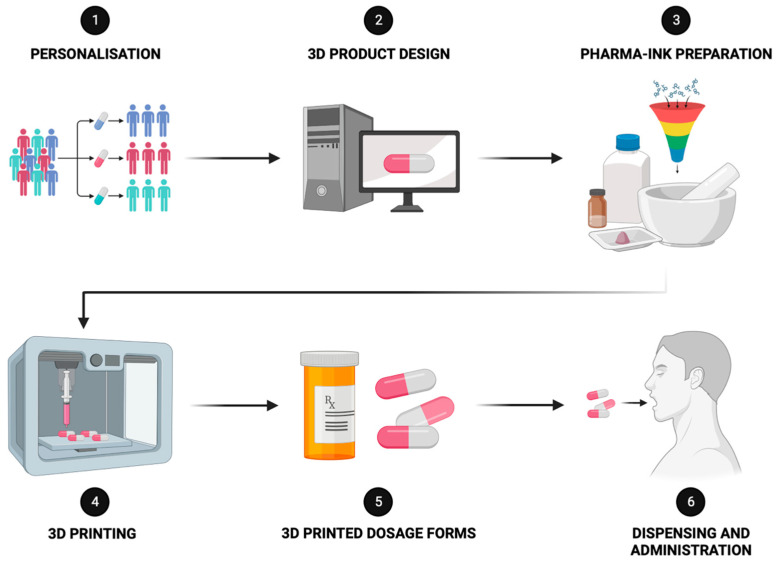
Schematic diagram of the steps included in the active ingredients’ 3D printing process.

**Figure 5 nutrients-16-00061-f005:**
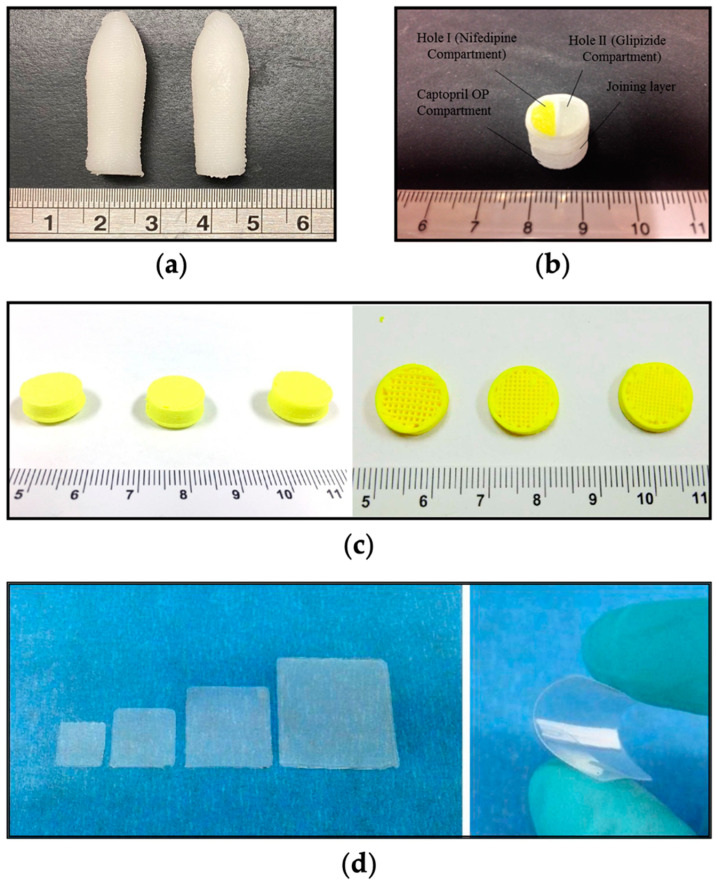
Pharmaceutical products created using SSE 3D printing: (**a**) a multi-drug loaded suppository [98]; (**b**) multi-active tablet containing captopril, nifedipine, and glipizide [88]; (**c**) (left) 3D printed gastro-floating tablets with different infilling percentage and (right) their respective cross-sections showing (from left to right) 30%, 50%, and 70% infill percentages [104]; and (**d**) flexible warfarin-loaded films [91]. All figures were reprinted with permission from their original sources.

**Figure 6 nutrients-16-00061-f006:**
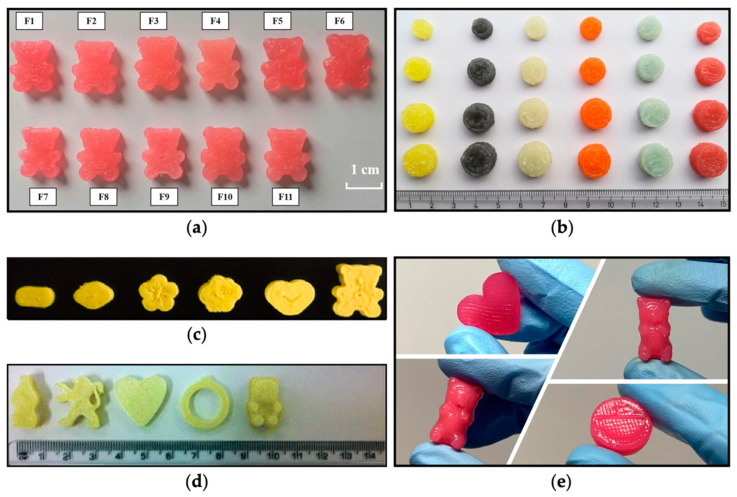
Advancements in chewable formulations: (**a**) chewable tablets shaped like gummy bears containing propranolol hydrochloride: F1–F11 formulations contain different amounts of gelatin and carrageenan [129]; (**b**) chewable isoleucine printlets offered in various flavors, colors, and doses [74]; (**c**) chewable tablets loaded with amlodipine besylate [120]; (**d**) chewable formulations containing indomethacin [121]; (**e**) gummies of different shapes containing ranitidine [122]. All figures were reprinted with permission from their original sources.

## Data Availability

No new data were created in this study. Data sharing is not applicable to this article.

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
