# Peer review of "3D Printing of Dietary Products for the Management of Inborn Errors of Intermediary Metabolism in Pediatric Populations"

_nutrients, 2023, doi:10.3390/nu16010061_

Round 1

Reviewer 1 Report

Comments and Suggestions for Authors

Thank you for the opportunity to review this interesting paper. I had the following comments/suggestions:

1. Section 3.1 - The authors do not mention the most commonly encountered monitoring device in metabolic disorders, continuous glucose monitors for patients with glycogen storage diseases. It would be good to mention these.

2. Overall I found most of the review complete and comprehensive except for the section "3.2.1 Precision medicine for children: SSE 3D printing's tailored solutions". I understand there are likely very few published studies on this technology in IEiMs but it would benefit the review for there to be more of a survey previous trials of 3D printed products in pediatric populations in general. I found this section very scant in relation to the overall goal of the paper and the rest of the information presented.

3. I think there are likely additional challenges to the implementation of this technology into the care of metabolic diseases than simply regulatory issues and QA/QC. What is the cost of this technology like? How applicable is this to metabolic disorders given than there are only a few instances where a supplemental nutrient is utilized? Do the authors anticipate this to be expanded to include printing of medications like benzoate/buphenyl? What are the potential effects of other components of the 3D printed products like fillers/binders?

4. This technology sounds most promising for older patients who are reasonably functional from a neurocognitive perspective and relatively mildly affected. How can this technology be utilized for younger children or those with limited ability to chew/swallow normally? This is likely where dissolvable films would come in handy but it would be nice for the authors to posit some views on this or at least have a section on potential applications. For example, what are the disorders the authors think are most amenable to this technology (perhaps a table with these could be inserted)?

Author Response

Reviewer #1:

Section 3.1 - The authors do not mention the most commonly encountered monitoring device in metabolic disorders, continuous glucose monitors for patients with glycogen storage diseases. It would be good to mention these.

We have now updated the manuscript to include continuous glucose monitors for patients with glycogen storage diseases in Section 3.1.

Overall I found most of the review complete and comprehensive except for the section "3.2.1 Precision medicine for children: SSE 3D printing's tailored solutions". I understand there are likely very few published studies on this technology in IEiMs but it would benefit the review for there to be more of a survey previous trials of 3D printed products in pediatric populations in general. I found this section very scant in relation to the overall goal of the paper and the rest of the information presented.

As mentioned by the reviewer, there is a limited number of clinical trials in children or in general population using 3D printing technologies. We had already included the only clinical study, which focused on MSUD. In addition, in the previous section, we generally discussed SSE and provided examples of multiple dosage forms that could be printed using this technology.

To strengthen this section, we have included more references to highlight the potential use of SSE 3D printed formulation for paediatric patients. The examples show different formulations containing a wide range of excipients and focus on paediatric treatments. They include the use of multiple excipients and/or drugs as well as formulations used to conceal drugs in children food during hospitalisation, offering the proper diet whilst improving adherence to treatment.

I think there are likely additional challenges to the implementation of this technology into the care of metabolic diseases than simply regulatory issues and QA/QC. What is the cost of this technology like? How applicable is this to metabolic disorders given than there are only a few instances where a supplemental nutrient is utilized? Do the authors anticipate this to be expanded to include printing of medications like benzoate/buphenyl? What are the potential effects of other components of the 3D printed products like fillers/binders?

This indeed a good point. However, the cost depends on several factors. As a starter, there are several 3D printers available on the market and the price of pharmaceutical 3D printers is affordable, especially when compared to tabletting machines, which are not considered to be cheap at all.

Similarly, the starting materials used for printing are the same as excipients commonly used in conventional tablet manufacturing. Therefore, there is no additional costs imposed. Moreover, for the majority of developed formulations, no additional post-printing steps are required. Similarly, no additional costs are associated with pre-printing steps since SSE pharma-inks are generally prepared by mixing the materials (both excipients and drug/s) under stirring and heating at low temperatures. The final paste or gel is loaded into a disposable syringe and cooled at room temperature to solidify the formulation before printing. In fact, once could argue that tabletting and capsule filling would impose additional costs as granulation is often needed. The information has now been included in the manuscript (Section 4).

With regards to its application in metabolic disorders, whilst there are only a few instances where a supplemental nutrient is utilised, in some patients, especially paediatric patients, small differences in the dosing could comprise the treatment outcomes.

3D printing has been explored for a wide array of drug products, including monoclonal antibodies. Thus, there is a possibility that its use could expand to include benzoate/buphenyl but whether these APIs remain stable following printing is another matter that requires further investigation and printing optimisation. It will also depend on the printing technology and formulation used.

Finally, the impacts of different excipients, including fillers and binders, on 3D printed products have been extensively researched, yielding varied outcomes in individual cases. Similar to tabletting or other pharmaceutical manufacturing methods, these effects are contingent upon factors like the included percentage, formulation, and the specific printing technology employed.

This technology sounds most promising for older patients who are reasonably functional from a neurocognitive perspective and relatively mildly affected. How can this technology be utilized for younger children or those with limited ability to chew/swallow normally? This is likely where dissolvable films would come in handy but it would be nice for the authors to posit some views on this or at least have a section on potential applications. For example, what are the disorders the authors think are most amenable to this technology (perhaps a table with these could be inserted)?

We appreciate the reviewer’s comment. However, we believe that chewable formulations are equally suitable for paediatric and geriatric patients.

Herein, the focus of this review is on paediatric patients since the clinical manifestations of inherited metabolic disorders start at an early age and children are generally known for their low adherence to treatment and for their difficulty or inability to swallow certain formulations. Not to mention that children have stricter and lower dosing requirements compared to adults, where dosages might need to be manipulated to acquire particular doses. SSE offers a means of dose personalisation, making it possible to produce age-appropriate formulations that meet the patients’ individual preferences (e.g., specific flavour, dosage form, or texture) (Section 3.2.1).

If the patient cannot chew the formulation, an orodispersible printlet or film can be prepared.

Regarding disorders that are the most amenable to SSE, as mentioned in previous comments, any drug or supplement can be easily added to the formulation. Thus, the applications are limitless. Thus, there are no specific disorders that are most amenable to this technology. This is why we provide examples of different disorders for which it has been explored.

Reviewer 2 Report

Comments and Suggestions for Authors

I was interested to read this article on new technologies to help with the dietary management of patients suffering from hereditary metabolic diseases. The article is detailed and the illustrations are of good quality. I would like to suggest a few comments/corrections for the first parts of the article.

Introduction: page 2: first lines "« Given their genetic origin, IMDs are relatively rare, typically manifesting in children who display early symptoms, underscoring the critical importance of prompt diagnosis for timely intervention and the prevention of metabolic or cardiac complications”: why only cardiac consequences. There can be multiple consequences: neurological, renal, etc.

2.1.1. substrate reduction therapyOf course, this type of treatment has been proposed for lysosomal diseases, but for this review based essentially on inborn errors of intermediary metabolism (IEiM), it will be interesting to consider the case of NTBC treatment in tyrosinemia type 1.

2.1.3. Scavenger therapy  :  I don’t understand the last paragraph : “In other IMD, such as lysosomal disorders, SRT aims to decrease the amount of substrate synthesised to a level at which it can be effectively cleared by the defected enzyme, thus restoring metabolic homeostasis. This therapy acts by blocking the production of substrate [17]. » it is not scavenger therapy. Probably a mistake

2.2.1. Cofactor supplementation : BH4 in phenylketonuria is also another example, probable more frequent than MSUD.

2.2.3. Dietary supplementation : perhaps, it will interesting to discuss other major dietary supplementations like in glycogen storage diseases or MCT in long-chain fatty acids disorders : sometimes very difficult diet.

Page 9 : what is “SSE” full name (only in the abstract)

General comment : is it possible to give anidea of the price of these new technologies ?

Author Response

Reviewer #2:

I was interested to read this article on new technologies to help with the dietary management of patients suffering from hereditary metabolic diseases. The article is detailed and the illustrations are of good quality. I would like to suggest a few comments/corrections for the first parts of the article.

We thank the reviewer for her/his feedback.

Introduction: page 2: first lines "« Given their genetic origin, IMDs are relatively rare, typically manifesting in children who display early symptoms, underscoring the critical importance of prompt diagnosis for timely intervention and the prevention of metabolic or cardiac complications”: why only cardiac consequences. There can be multiple consequences: neurological, renal, etc.

We have now replaced the term cardiac consequences with “severe multisystemic consequences”.

2.1.1. substrate reduction therapy : Of course, this type of treatment has been proposed for lysosomal diseases, but for this review based essentially on inborn errors of intermediary metabolism (IEiM), it will be interesting to consider the case of NTBC treatment in tyrosinemia type 1.

We would like to thank the reviewer for his/suggestion suggestion. We have now updated the manuscript to include this in Section 2.1.1.

2.1.3. Scavenger therapy  :  I don’t understand the last paragraph : “In other IMD, such as lysosomal disorders, SRT aims to decrease the amount of substrate synthesised to a level at which it can be effectively cleared by the defected enzyme, thus restoring metabolic homeostasis. This therapy acts by blocking the production of substrate [17]. » it is not scavenger therapy. Probably a mistake

We regret this oversight. We have now excluded this sentence from the manuscript.

2.2.1. Cofactor supplementation : BH4 in phenylketonuria is also another example, probable more frequent than MSUD.

This example has now been included in Section 2.2.1.

2.2.3. Dietary supplementation : perhaps, it will interesting to discuss other major dietary supplementations like in glycogen storage diseases or MCT in long-chain fatty acids disorders : sometimes very difficult diet.

We have now updated the manuscript to include this in Section 2.2.3.

Page 9 : what is “SSE” full name (only in the abstract)

We thank the reviewer for pointing this out. We have now defined the abbreviation before using it.

General comment : is it possible to give an idea of the price of these new technologies?

This indeed a good point. However, the cost depends on several factors. As a starter, there are several 3D printers available on the market and the price of pharmaceutical 3D printers is affordable, especially when compared to tabletting machines, which are not considered to be cheap at all.

Similarly, the starting materials used for printing are the same as excipients commonly used in conventional tablet manufacturing. Therefore, there is no additional costs imposed. Moreover, for the majority of developed formulations, no additional post-printing steps are required. Similarly, no additional costs are associated with pre-printing steps since SSE pharma-inks are generally prepared by mixing the materials (both excipients and drug/s) under stirring and heating at low temperatures. The final paste or gel is loaded into a disposable syringe and cooled at room temperature to solidify the formulation before printing. In fact, once could argue that tabletting and capsule filling would impose additional costs as granulation is often needed. The information has now been included in the manuscript (Section 4).

Round 2

Reviewer 1 Report

Comments and Suggestions for Authors

Thank you for the opportunity to review this manuscript once again. The authors have clearly taken my previous comments into consideration and the manuscript is much improved.